# Knowledge, attitude, and practice regarding venous thromboembolism prophylaxis: A multicenter cross-sectional study of medical staff in Guinea

Xinnong Liu[1], Soriba Naby Camara[2], Mamady Diakite[3], Denis Bernard Raiche[4], Zhujiazi Zhang[5]*

1 Department of Vascular Surgery, Beijing Tiantan Hospital, Capital Medical University, Beijing, China, 2 Department of General surgery, China-Guinea Friendship Hospital, Conakry, Guinea, 3 Department of hematology, Ignace Deen Hospital, Conakry, Guinea, 4 General administrator, Donka University Hospital, Conakry, Guinea, 5 Department of Immunization and Prevention, Beijing Center for Disease Prevention and Control, Beijing, China

* jiazi8515@163.com

## Abstract

### Objectives

To investigate the awareness of medical staff regarding venous thromboembolism (VTE) prophylaxis in Guinea.

### Methods

The survey was completed from June 1, 2023 to August 1, 2023 through filling out self-designed questionnaire including four parts containing demographic data, knowledge, attitude, and practice regarding VTE prophylaxis. Cronbach's alpha values were used to analyze the internal consistency of the questionnaire. The results were analyzed using chi-square tests at a 95% significance level.

### Results

Of the 245 medical staff invited to participate in the survey, 211 (86.1%) responded. Cronbach's alpha value of the questionnaire was 0.92. The overall correct response rate for knowledge was 61.5±11.7%, and there were no significant differences between hospitals, sexes, professions, educational levels, departments, and working years (P > 0.05). The overall affirmative response rates for attitude and practice were 65.3%±18.4% and 74.8±13.4%, respectively. The affirmative rate of nurses was higher than that of clinicians in the aspects of attitude (69.51±20.2% vs. 63.0±18.1%) and practice (82.1±16.9% vs. 70.4±10.8%); however, no significant difference was found (P > 0.05).

**Data availability statement:** The datasets generated and/or analyzed during the current study are available for all researchers. They can visit the S1 Appendix and S2 Appendix directly.

**Funding:** This research was supported by the "Belt and Road Initiative" International Health Cooperation Project and the WHO Cooperation Project (2022-2023 and 2024-2025).

**Competing interests:** The authors have declared that no competing interests exist.

## Conclusions

The knowledge level, attitude, and practice regarding VTE prophylaxis among medical staff in Guinea were generally poor. We suggest that medical institutions provide appropriate VTE prophylaxis-related trainings.

## 1. Introduction

Venous thromboembolism (VTE), which includes deep venous thrombosis (DVT) and pulmonary embolism (PE), is a major public health issue worldwide and is estimated to be the third most common cardiovascular event in Western countries [1,2]. In the United States, there were an average of 547,596 adult hospitalizations with a diagnosis of VTE each year from 2007 to 2009 in a population of 301–307 million [3]. In recent decades, the prevalence of VTE has increased steadily in developing countries. A multicenter retrospective study in China found that the VTE-related hospitalization rates increased from 3.2 per 100,000 to 17.5 per 100,000 from 2007 to 2016 [4]. Despite the gradual increase in VTE disease burden, there is a general lack of awareness of VTE prevention in different regions [5].

Guinea is a West African country with poor healthcare and a double burden of communicable and non-communicable diseases [6,7]. VTE is one of the leading causes of death for hospitalized patients and is considered a preventable disease if optimal prophylactic strategies are employed [8]. Medical staff play a key role in preventing VTE by assessing VTE risk and providing appropriate prophylactic measures. Research has shown that the knowledge and attitude of clinicians regarding VTE can affect the efficacy of VTE prophylaxis [9]. Despite the importance of preventing VTE among hospitalized patients, no research has been conducted on the knowledge, attitudes, and practices of medical staff regarding VTE prophylaxis in Guinea.

Therefore, in this survey, we sought to assess the knowledge, attitude, and practice of medical staff regarding VTE prophylaxis at the national hospital level in Guinea, which will contribute to providing suggestions for VTE-related training, health policy development, and in-hospital VTE prevention and treatment in Guinea.

## 2. Methods

### 2.1. Study design

A cross-sectional survey was conducted among medical staff in national hospitals in Guinea to evaluate the knowledge, attitude, and practice of medical staff regarding venous thromboembolism prophylaxis from June 1, 2023, to August 1, 2023. Guinea has three national hospitals: China–Guinea Friendship Hospital, Ignace Deen Hospital, and Donka University Hospital. Inclusion criteria: The participants included all the clinicians, nurses, and pharmacists from all the clinical departments including internal medicine (Cardiology, Neurology, Acupuncture, Gastroenterology, Endocrinology, Respiratory medicine, and Nephrology), surgery (Neurosurgery, General surgery, Urology, Thoracic surgery, and Operating room),

Intensive Care Unit (ICU), and Emergencies. Exclusion criteria: Administrators, Laboratory Physicians, Imaging Physicians, and Logistics Department staff. This study was approved by the Ethics Review Committee of the China–Guinea Friendship Hospital. Written informed consent was not required because the survey of medical staff was anonymous and had minimal risk.

## 2.2. Questionnaire design

The questionnaire was designed according to previous study and was evaluated and revised with respect to methodology and content by eight experts in nursing, medical, and surgical fields [10, 11]. The final validated questionnaire consisted of four parts with 83 items: demographic data (seven items) and knowledge (63 items), attitude (nine items), and practices regarding VTE prophylaxis (four items). The demographic data collected included respondents' occupation, age, department, years of work, educational level, and professional title. Questions about knowledge included five topics with 63 items consisting of basic knowledge, risk assessment, basic prophylaxis, physical prophylaxis, and pharmacological prophylaxis. Questions about attitude and practice included the following possible responses: "strongly agree," "agree," "neutral," "disagree," and "strongly disagree." Answers defined as "affirmative" included responses of "agree" and "strongly agree." All the survey data were entered and checked twice to ensure consistency and accuracy. Cronbach's alpha values were used to analyze the internal consistency of the questionnaire.

## 2.3. Survey procedure

The questionnaire was administered through an on-site survey. A cover letter was sent to all participants to explain the purpose of the study. All the questionnaires were completed under the supervision of an investigator. All participants were asked to answer the questions objectively and honestly. The following two steps were mandatory to complete this survey: 1) participants were asked to report their demographic characteristics, and 2) participants had to complete all questions regarding knowledge, attitude, and practice. Questionnaires with incomplete information were excluded.

## 2.4. Statistical analysis

Descriptive statistics were used to present demographic data, the correct response rate for participants' knowledge, and the rates of affirmative responses for VTE prophylaxis attitude and practice; these are expressed as percentages or as mean±standard deviations (SD). Differences in average correct response rates were compared through analysis of variance, and differences in affirmative response rates were compared using the chi-squared test. A p-value of $\leq 0.05$ (two-tailed) was considered statistically significant. All statistical analyses were performed using SPSS software (version 17.0; SPSS Inc., Chicago, Illinois).

## 2.5. Inclusivity in global research

Additional information regarding the ethical, cultural, and scientific considerations specific to inclusivity in global research is included in the S3 Appendix.

## 3. Results

### 3.1. General characterisitcs of participants

A total of 245 questionnaires were distributed, of which 211 were included in the analysis, with a response rate of 86.1%. Cronbach's alpha value of the questionnaire was 0.92. Of the 211 returned questionnaires, 65 were received from China–Guinea Friendship Hospital, 133 from Ignace Deen Hospital, and 13 from Donka University Hospital. There were 125 males and 86 females, with 108 clinicians, 98 nurses, and five others (four pharmacists and one laboratory technician). The average age of all participants was 35.6±10.1 years, and the average working years was 8.4±7.6 years.

## 3.2. Knowledge toward VTE prophylaxis

The overall correct response rate was 61.5±11.7%. There were no significant differences in the accuracy rates among hospitals, sexes, professions, educational levels, departments, and working years (All $P > 0.05$, as shown in Table 1). The correct response rates for basic knowledge, risk assessment, basic prophylaxis, physical prophylaxis, and pharmacological prophylaxis were 70.8±14.4%, 65.0±16.5%, 72.3±22.3%, 44.7±19.8%, and 45.7±22.5%, respectively, with significant differences ($P < 0.001$).

## 3.3. Attitude toward VTE prophylaxis

The overall affirmative response rate for attitude was 65.3±18.4%, and there was no significant difference between clinicians and nurses (63.0±18.1% vs. 69.5±20.2%, $P = 0.330$). Compared with those of clinicians, nurses had significantly higher affirmative response rates for Q7, Q9.3, and Q9.4 ($P < 0.001$) (Table 2). As for the attitude toward VTE assessment (Q1), treatment (Q2), and training (Q4), subgroup analysis based on different departments showed that the affirmative rates of emergency/ICU (Q1: 97.9%, Q2: 89.6%, Q4: 97.9%) were higher than those in internal medicine (Q1: 84.3%, Q2: 77.5%, Q4: 85.4%) and surgery (Q1: 93.9%, Q2: 90.9%, Q4: 95.5%) ($P = 0.023$ for Q1, $P = 0.049$ for Q2 and $P = 0.021$ for Q4).

**Table 1. Knowledge regarding VTE prophylaxis of medical staff in Guinea.**

| Characteristic | Number of participants, n (%) | Average correct response rate±SD (%) | *P*-value |
|---|---|---|---|
| Total | 211 (100) | 61.5±11.7 | |
| Hospital | | | |
| China-Guinea Friendship Hospital | 65 (30.8) | 64.0±9.9 | 0.060 |
| Ignace Deen Hospital | 133 (63.0) | 60.0±12.2 | |
| Donka University Hospital | 13 (6.2) | 64.0±13.4 | |
| Sex | | | |
| Male | 125 (59.2) | 61.8±12.6 | 0.644 |
| Female | 86 (40.8) | 61.0±10.2 | |
| Profession | | | |
| Clinician | 108 (51.2) | 60.8±13.5 | 0.127 |
| Nurse | 98 (46.4) | 62.6±9.3 | |
| Others | 5 (2.4) | 52.7±9.1 | |
| Educational level | | | |
| High school and below | 95 (45.0) | 61.5±10.3 | 0.877 |
| Bachelor's degree | 14 (6.6) | 59.6±12.5 | |
| Master's degree | 97 (46.0) | 61.6±13.1 | |
| Others | 5 (2.4) | 64.4±7.3 | |
| Department | | | |
| Internal medicine | 89 (42.2) | 60.8±12.4 | 0.770 |
| Surgery | 66 (31.3) | 62.2±9.8 | |
| Emergency/ICU | 48 (22.7) | 61.3±13.0 | |
| Others | 8 (3.8) | 64.5±11.1 | |
| Years of work | | | |
| <5 | 74 (35.1) | 61.0±11.8 | 0.673 |
| 5-10 | 76 (36.0) | 60.8±13.7 | |
| 11-20 | 44 (20.9) | 63.3±8.6 | |
| >20 | 17 (8.1) | 62.0±8.5 | |

**Table 2. Attitude regarding VTE prophylaxis of medical staff in Guinea.**

| Item | Affirmative response, n (%) (N = 211) | Clinician, n (%) (N = 108) | Nurse, n (%) (N = 98) | *P*-value* |
|---|---|---|---|---|
| Q1. VTE risk must be assessed in hospitalized patients. | 191 (90.5) | 96 (88.9) | 91 (92.9) | 0.348 |
| Q 2. A medical specialist must provide therapy to patients with VTE. | 178 (84.4) | 90 (83.3) | 84 (85.7) | 0.702 |
| Q 3. A multidisciplinary team must provide therapy to patients with VTE. | 163 (77.3) | 81 (75.0) | 78 (79.6) | 0.507 |
| Q 4. Staff must be trained regularly regarding VTE prophylaxis. | 194 (91.9) | 97 (89.8) | 94 (95.9) | 0.112 |
| Q 5. VTE prophylaxis can improve the quality of medical care. | 181 (85.8) | 91 (84.3) | 88 (89.8) | 0.303 |
| Q 6. Your medical division encourages you to learn more about VTE prophylaxis. | 161 (76.3) | 78 (72.2) | 80 (81.6) | 0.138 |
| Q 7. Your hospital pays a great deal of attention to VTE prophylaxis. | 142 (67.3) | 58 (53.7) | 81 (82.7) | 0.000 |
| Q 8. What are your concerns regarding VTE prophylaxis? | | | | |
| 8.1 Financial penalty when the patient cannot be treated with VTE prophylaxis | 94 (44.5) | 49 (45.4) | 45 (45.9) | 1.000 |
| 8.2 Increased workload | 72 (34.1) | 36 (33.3) | 36 (36.7) | 0.662 |
| 8.3 Increased medical cost | 106 (50.2) | 62 (57.4) | 43 (43.9) | 0.069 |
| 8.4 Extended hospital stay | 108 (51.2) | 59 (54.6) | 48 (49.0) | 0.485 |
| 8.5 Exacerbation of doctor-patient conflicts | 72 (34.1) | 33 (30.6) | 38 (38.8) | 0.242 |
| Q 9. What are the difficulties involved in VTE prophylaxis? | | | | |
| 9.1 Staff's knowledge and participation | 154 (73.0) | 79 (73.2) | 75 (76.5) | 0.632 |
| 9.2 Patient compliance | 122 (57.8) | 58 (53.7) | 64 (65.3) | 0.118 |
| 9.3 Cooperation among different departments | 153 (72.5) | 68 (63.0) | 84 (85.7) | 0.000 |
| 9.4 Ability of medical specialist to treat VTE | 129 (61.1) | 56 (51.9) | 73 (74.5) | 0.001 |
| 9.5 Medical cost | 123 (58.3) | 66 (61.1) | 56 (57.1) | 0.574 |

*Compared with clinicians and nurses.

### 3.4. Practice toward VTE prophylaxis

The overall affirmative response rate for practice was 74.8 ± 13.4%, and there was no significant difference between clinicians and nurses (70.4 ± 10.8% vs. 82.1 ± 16.9%, $P = 0.286$). Compared with those of clinicians, nurses had significantly higher affirmative response rates for VTE assessment and patient education (Q1-Q3, $P < 0.001$). Clinicians and nurses had lower competence in VTE assessment (Q4) (Table 3).

## 4. Discussion

The present survey was the first to estimate the knowledge, attitudes, and practices of medical staff regarding VTE prophylaxis in Guinea. The results showed that 1) the overall knowledge level regarding VTE prophylaxis was poor among

**Table 3. Practice regarding VTE prophylaxis of medical staff in Guinea.**

| Item | Affirmative response, n (%) (N = 211) | Clinician, n (%) (N = 108) | Nurse, n (%) (N = 98) | *P*-value* |
|---|---|---|---|---|
| Q 1. You always assess VTE risk in hospitalized patients. | 169 (80.1) | 79 (73.2) | 89 (90.8) | 0.001 |
| Q 2. You always provide health education regarding VTE prophylaxis for hospitalized patients. | 163 (77.3) | 77 (71.3) | 85 (86.7) | 0.010 |
| Q 3. You can offer advice to patients with VTE. | 182 (86.3) | 88 (81.5) | 92 (93.9) | 0.011 |
| Q 4. You understand and have mastered the VTE risk assessment scales. | 117 (55.5) | 60 (55.6) | 56 (57.1) | 0.888 |

*Compared with doctors and nurses.

clinicians and nurses; 2) the overall attitude toward VTE prophylaxis was low, while the overall attitude of nurses toward to VTE prophylaxis was more positive than that of clinicians; 3) the affirmative response rate regarding practices was relatively higher, especially among nurses.

The knowledge of medical staff regarding VTE prophylaxis can affect the efficacy of these practices, with poor knowledge about VTE prophylaxis leading to a lack of standardization and potentially increasing the occurrence of VTE among hospitalized patients [12]. Our present survey found that the knowledge level regarding VTE prophylaxis was poor, and this result was not affected by hospital, sex, profession, educational level, department, and working years. Among these five topics, the correct rates for physical and pharmacological prophylaxis were significantly lower than those for basic knowledge, risk assessment, and basic prophylaxis. Therefore, we conclude that it is imperative for medical staff to improve their knowledge of VTE prophylaxis, especially for physical and pharmacological prophylaxis, either through self-study or by participating in related training on VTE prophylaxis.

Regarding attitudes toward VTE prophylaxis, although there was no significant difference in the overall affirmative response rate between clinicians and nurses, we found a positive attitude toward VTE prophylaxis among nurses. A subgroup analysis based on different departments showed that the surgery and ICU/emergency departments had a more positive attitude in terms of VTE assessment, treatment, and medical staff training than that of the internal medicine department. VTE is more common in surgical and critically ill patients, which maybe explain medical staff in the surgery or ICU have a more positive attitude toward VTE prophylaxis. A previous study has shown that VTE prophylaxis has become a standard of preventive measure in the ICU and surgical departments of developed and developing countries [13,14].

Compared with that of knowledge and attitude, the rate of affirmative responses to practice was relatively higher and the affirmative responses of the medical staff were higher than that performed in Chinese study [10]. The affirmative responses of clinicians were lower than those of nurses. Our results also showed that medical staff, including clinicians and nurses, have not completely mastered VTE prophylaxis, which may have affected overall practices. Our findings suggest that the medical staff, especially clinicians, should be encouraged to prioritize VTE prophylaxis. To date, multiple validated VTE risk assessment scales exist, and the medical staff should choose the appropriate one based on their area of specialization [15,16].

This study had several limitations. The survey was administered at national hospitals, and these findings may not be generalizable to other hospitals. However, the included hospitals were all national hospitals that could represent the highest level in Guinea. One hospital received very few responses, which may have biased the results. In addition, although the questionnaire was designed by a research team, it may still contain flaws that could have affected our findings. Finally, Multiple VTE risk assessment scales exist, while our present study did not specify a particular scale, future studies may choose the appropriate scale according to specialty.

In conclusion, the medical staff's knowledge level, attitude, and practice regarding VTE prophylaxis were generally poor. Based on these findings, we suggest that medical institutions provide relative training on VTE prophylaxis for medical staff in Guinea. However, VTE prophylaxis is a multidisciplinary effort that also requires the involvement and coordination of hospital administrators.

## Supporting information

**S1 Appendix:  The questionnaire used for this study.**
(DOCX)

**S2 Appendix:  The original data of our questionnaire.**
(XLSX)

**S3 Appendix:  The statement of inclusivity in global research.**
(DOCX)

## Author contributions

**Conceptualization:** Xinnong Liu, Zhujiazi Zhang.

**Data curation:** Xinnong Liu, Soriba Naby Camara, Mamady Diakite, Denis Bernard Raiche.

**Formal analysis:** Mamady Diakite, Zhujiazi Zhang.

**Funding acquisition:** Zhujiazi Zhang.

**Investigation:** Denis Bernard Raiche.

**Methodology:** Soriba Naby Camara, Denis Bernard Raiche.

**Project administration:** Soriba Naby Camara, Mamady Diakite, Denis Bernard Raiche, Zhujiazi Zhang.

**Resources:** Mamady Diakite.

**Software:** Zhujiazi Zhang.

**Supervision:** Soriba Naby Camara, Zhujiazi Zhang.

**Validation:** Mamady Diakite, Denis Bernard Raiche.

**Writing – original draft:** Mamady Diakite.

**Writing – review & editing:** Xinnong Liu, Zhujiazi Zhang.

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
