## [Decision Letter · Decision Letter 0]

20 Oct 2024

PONE-D-24-29660Knowledge, attitude, and practice regarding venous thromboembolism prophylaxis: A multicenter cross-sectional study of medical staff in GuineaPLOS ONE

Dear Dr. Liu,

Thank you for submitting your manuscript to PLOS ONE. After careful consideration, we feel that it has merit but does not fully meet PLOS ONE’s publication criteria as it currently stands. Therefore, we invite you to submit a revised version of the manuscript that addresses the points raised during the review process.

We look forward to receiving your revised manuscript.

Kind regards,

Mohammed Hussain Abutaleb, PhD

Academic Editor

PLOS ONE

Journal Requirements:

https://journals.lww.com/md-journal/fulltext/2021/12100/knowledge,_attitudes,_and_practices_regarding.27.aspx

In your revision ensure you cite all your sources (including your own works), and quote or rephrase any duplicated text outside the methods section. Further consideration is dependent on these concerns being addressed.

4. Please note that funding information should not appear in any section or other areas of your manuscript. We will only publish funding information present in the Funding Statement section of the online submission form. Please remove any funding-related text from the manuscript.

5. We note that the grant information you provided in the ‘Funding Information’ and ‘Financial Disclosure’ sections do not match. When you resubmit, please ensure that you provide the correct grant numbers for the awards you received for your study in the ‘Funding Information’ section.

6. Thank you for stating the following financial disclosure: “This research was supported by the “Belt and Road Initiative” International Health Cooperation Project and the WHO Cooperation Project (2022-2023 and 2024-2025).”

7. Please provide a complete Data Availability Statement in the submission form, ensuring you include all necessary access information or a reason for why you are unable to make your data freely accessible. If your research concerns only data provided within your submission, please write "All data are in the manuscript and/or supporting information files" as your Data Availability Statement.

8. When completing the data availability statement of the submission form, you indicated that you will make your data available on acceptance. We strongly recommend all authors decide on a data sharing plan before acceptance, as the process can be lengthy and hold up publication timelines. Please note that, though access restrictions are acceptable now, your entire data will need to be made freely accessible if your manuscript is accepted for publication. This policy applies to all data except where public deposition would breach compliance with the protocol approved by your research ethics board. If you are unable to adhere to our open data policy, please kindly revise your statement to explain your reasoning and we will seek the editor's input on an exemption. Please be assured that, once you have provided your new statement, the assessment of your exemption will not hold up the peer review process.

Reviewers' comments:

Reviewer's Responses to Questions

**Comments to the Author**

1. Is the manuscript technically sound, and do the data support the conclusions?

Reviewer #1: Partly

Reviewer #2: Yes

2. Has the statistical analysis been performed appropriately and rigorously? 

Reviewer #1: No

Reviewer #2: No

3. Have the authors made all data underlying the findings in their manuscript fully available?

Reviewer #1: No

Reviewer #2: Yes

4. Is the manuscript presented in an intelligible fashion and written in standard English?

Reviewer #1: Yes

Reviewer #2: Yes

5. Review Comments to the Author

Reviewer #1: Article review PONE-D-24-29660

For Editors: Thank you for the opportunity to review this manuscript. Though I have some major concerns on the methods and results, I encourage the authors to address the concerns and resubmit for publication. In a brief PubMed search, I was unable to find any published material regarding VTE in Guinea. As developing nations increase their risk factors for VTE, it is important that their healthcare systems facilitate VTE prevention education for providers and patients. Thus, this work is likely to be an important step in the process.

Available comments for Authors:

Summary: The authors describe an important analysis of healthcare provider knowledge and attitudes regarding VTE prophylaxis in Guinea. This work is innovative in this population and a better understanding can help hospitals, universities, and other agencies advocate for increased awareness and education to address barriers and gaps. Though the manuscript is well-written, there are several areas that require significant revision of the methods, statistical results, conclusions.

Data Availability: Authors state data is available but did not describe where the data may be found.

Abstract:

• The objective is reporting the awareness of medical staff regarding VTE prophylaxis using a survey examining the demographic data, knowledge, attitude, and practice regarding VTE prophylaxis.

o How this leads to the ability of the authors to “provide suggestions for improvement in VTE prevention and treatment” is unclear and is never addressed further in the manuscript.

Introduction:

• Line 43: Reference 3 is 25 years out of date. In discussions of disease burden a more recent reference would be preferred. For example, the author of Reference 2 published an estimate of US VTE burden in 2021. doi: 10.1055/s-0040-1722189.

• Line 48: Reference 5 does not apply to the sentence, nor does it apply to the previous sentence to explain the rise in VTE-related hospitalizations in China. A more applicable reference is warranted.

• Line 49-50: Refence 6 discusses gender related risk factors for cardiovascular disease among several sub-Saharan African nations. As this is just one non-communicable disease impacting Guinea, it does support the sentence of “poor healthcare and a double burden of communicable and non-communicable diseases”. Suggest adding references for support, for example: the WHO data on Global health estimates by country.

• Line 50-52: While it is well known that proper prophylactic protocols have the potential to prevent VTE among hospitalized patients, Reference 7 is a review of risk factors not prophylaxis. Suggest a more targeted reference. Also, the phrase “unintended death in hospitalized patients” implies there are “intended death in hospitalized patients”, different phrasing may be a reasonable consideration.

Methods:

• Overall comments: The authors well describe the population and analysis, though a few questions remain.

• 34 questionnaires were not returned. The authors do not describe the reasons for non-returned surveys or if the surveys were returned but incomplete and thus excluded.

• How was the survey validated in this population?

Results:

• Overall comments: The authors did not include the list of questions asked regarding VTE prophylaxis knowledge. References 9 and 10 link to previously published surveys that were adapted for this study, consisting of 21 and 68 knowledge questions respectively. Since the previous surveys consist of differing numbers of questions and the previous populations were both in China, it is questionable if the authors can establish the current survey as validated as no steps for validation in this population or the current number of knowledge questions are described. In addition, if the knowledge questions are similar in scope to the questions described in Reference 10, then the majority of questions are related to VTE risk factors, and the text of the manuscript should accurately reflect that fact. Statistics on correct knowledge responses and Table 1 are recommended to be reanalyzed by knowledge categories, i.e. signs and symptoms, risk factors, prophylaxis, etc.

• Table 1 education level: Correct the spelling of “Heigh” to High.

• Table 2 Attitude regarding VTE prophylaxis of medical staff in Guinea.

o These questions should NOT be combined to create a single overall affirmative response rate. Two of the questions are assessing the respondent’s assessment of their employers’ opinion of VTE prophylaxis; and two are multi part questions assessing concerns and difficulties where a negative response would be better assessed as a positive.

• Table 3. Practice regarding VTE prophylaxis of medical staff in Guinea.

o Is it standard practice in Guinea for nurses to conduct VTE risk assessment? The discussion should reflect that. In the US, VTE risk assessment is usually conducted by clinicians.

o Question 4 is very vague and the much lower affirmative response rates may reflect that vagueness. Which VTE risk assessment scales are being referenced? In the US there are many different VTE risk assessment scales in practice, some are for different populations (cancer, pregnancy, pediatric) or different settings (medical, surgical, trauma) or developed in house versus externally validated. It is unreasonable to require providers in different specialties to have mastered ALL VTE risk assessment scales.

 Given the above issue, it is also recommended that an overall affirmative response rate not be reported.

Discussion:

• Line 161: The authors discuss results that were not presented in the results section. But these are results I’m recommending they present, the breakout of topic areas for the knowledge questions.

• Line 168: “found a positive attitude toward VTE prophylaxis among nurses” Should the word “more” been included? Also, must present the sub-analysis data in the results.

• Line 171-174: Sentence needs adjustment to not make a causation statement.

• Line 174-176: Reference 12 is a review article and 13 occurs in Spain; neither address the statement as applied to “developing countries”.

• Line 178-181: Reference 14 presents the Padua Prediction Score and Reference 15 presents ASH’s guidelines for VTE among patients with cancer; at no time do either mention that conducting a risk assessment is mandatory for nurses but not clinicians. This is a disingenuous use of these references. If nurses conducting risk assessment is standard practice in Guinea then the authors should just state that fact.

• Line 181-183: See previous comment regarding the vagueness of the question regarding VTE risk assessment “score”.

Conclusion:

• Current conclusion is based on the current results. There may be a significant shift if authors undertake suggest revisions to methodology and analysis.

Reviewer #2: Xinnong Liu et al report on "Knowledge, attitude, and practice regarding venous thromboembolism prophylaxis: A

multicenter cross-sectional study of medical staff in Guinea". The authors report on data from filling out self-designed questionnaire including four parts composed of demographic data, knowledge, attitude, and practice regarding VTE prophylaxis. The authors analysed 211 (86.1%).staff member responders' questionaires. "Correct" respnses were found in a range of 65% up to 84%, which was considered poor by the authors and the somewhat simple conclusion was to train better.

The topic is timely and of interest.

Emerging countries are cathching up and should be supported in their endeavour to train the staff in thrombosis prophylaxis.

Issues that should be considered:

1) 65%-84% positive answers should be wighed more differentiated. The (vast) majority seem to be compliant with the defined endpoints and answers correctly.

2) Given the situation in a emerging country, the verdict should be primarily positive.

3) The direct comparision of a partner hospital in China and /or of the western world would be desirable.

4) Data from the literature could alternatively flow in for more direct comparisions.

5) Consider also that ideal situations of procedures in emerging countries could differ greatly from other countries in the western world.

6) One example: "A multidisciplinary team must provide therapy to patients with VTE". This is how it works in other countries, but the structure of command may differ in Guinea.

7) "must" in VTE prophylaxis may induce counter actions and negative feelings towards outside authorities.

8) "What arrives" actually to the patients in need of VTE prophylaxis and to the ones who should not get it would be an important addition to the study, data may be available.

9) If not, recent data from other countries surveys on VTE prophylaxis could be directly compared.

10) More constructive advices on how to improve and perhaps to support the quality-measures would be very welcome, since this may apply to many other countries.

6. PLOS authors have the option to publish the peer review history of their article (what does this mean? ). If published, this will include your full peer review and any attached files.

**Do you want your identity to be public for this peer review?** For information about this choice, including consent withdrawal, please see our Privacy Policy .

Reviewer #1: No

Reviewer #2: No

---

## [Author Response · Author response to Decision Letter 0]

6 Nov 2024

Reviewer's Responses to Questions

Reviewer #1: Article review PONE-D-24-29660

For Editors: Thank you for the opportunity to review this manuscript. Though I have some major concerns on the methods and results, I encourage the authors to address the concerns and resubmit for publication. In a brief PubMed search, I was unable to find any published material regarding VTE in Guinea. As developing nations increase their risk factors for VTE, it is important that their healthcare systems facilitate VTE prevention education for providers and patients. Thus, this work is likely to be an important step in the process.

Available comments for Authors:

Summary: The authors describe an important analysis of healthcare provider knowledge and attitudes regarding VTE prophylaxis in Guinea. This work is innovative in this population and a better understanding can help hospitals, universities, and other agencies advocate for increased awareness and education to address barriers and gaps. Though the manuscript is well-written, there are several areas that require significant revision of the methods, statistical results, conclusions.

Data Availability: Authors state data is available but did not describe where the data may be found.

Response: Thank you for your suggestions, we have unloaded the data and stated that in the part of data of our manuscript.

Abstract:

• The objective is reporting the awareness of medical staff regarding VTE prophylaxis using a survey examining the demographic data, knowledge, attitude, and practice regarding VTE prophylaxis.

o How this leads to the ability of the authors to “provide suggestions for improvement in VTE prevention and treatment” is unclear and is never addressed further in the manuscript.

Response: Thank you for your suggestions, our questionnaire included both knowledge of VTE prevention and aspects of VTE treatment. We agree with your suggestion and have made changes, thank you.

Introduction:

• Line 43: Reference 3 is 25 years out of date. In discussions of disease burden a more recent reference would be preferred. For example, the author of Reference 2 published an estimate of US VTE burden in 2021. doi: 10.1055/s-0040-1722189.

Response: We have replaced this reference according your suggestion, thank you very much.

• Line 48: Reference 5 does not apply to the sentence, nor does it apply to the previous sentence to explain the rise in VTE-related hospitalizations in China. A more applicable reference is warranted.

Response: Thank you for your suggestion, we have revised the sentence and replaced this reference according your suggestion, thank you very much.

• Line 49-50: Refence 6 discusses gender related risk factors for cardiovascular disease among several sub-Saharan African nations. As this is just one non-communicable disease impacting Guinea, it does support the sentence of “poor healthcare and a double burden of communicable and non-communicable diseases”. Suggest adding references for support, for example: the WHO data on Global health estimates by country.

Response: Thank you for your suggestion, we have added the reference that states the disease burden data from WHO, according your suggestion, thank you very much.

• Line 50-52: While it is well known that proper prophylactic protocols have the potential to prevent VTE among hospitalized patients, Reference 7 is a review of risk factors not prophylaxis. Suggest a more targeted reference. Also, the phrase “unintended death in hospitalized patients” implies there are “intended death in hospitalized patients”, different phrasing may be a reasonable consideration.

Response: Thank you for your suggestion, we have replaced the reference according your suggestion, thank you very much.

Methods:

• Overall comments: The authors well describe the population and analysis, though a few questions remain.

• 34 questionnaires were not returned. The authors do not describe the reasons for non-returned surveys or if the surveys were returned but incomplete and thus excluded.

• How was the survey validated in this population?

Response:

The 34 questionnaires included questionnaires with incomplete information and these questionnaires with incomplete information were excluded. Before performing our survey, our questionnaire was validated and revised by eight experts in nursing, medical, and surgical fields, which was described in method and result part of manuscript. Thank you for your suggestions.

Results:

• Overall comments: The authors did not include the list of questions asked regarding VTE prophylaxis knowledge. References 9 and 10 link to previously published surveys that were adapted for this study, consisting of 21 and 68 knowledge questions respectively. Since the previous surveys consist of differing numbers of questions and the previous populations were both in China, it is questionable if the authors can establish the current survey as validated as no steps for validation in this population or the current number of knowledge questions are described. In addition, if the knowledge questions are similar in scope to the questions described in Reference 10, then the majority of questions are related to VTE risk factors, and the text of the manuscript should accurately reflect that fact. Statistics on correct knowledge responses and Table 1 are recommended to be reanalyzed by knowledge categories, i.e. signs and symptoms, risk factors, prophylaxis, etc.

• Table 1 education level: Correct the spelling of “Heigh” to High.

Response:

Thank you for your suggestion, we have revised that according to your suggestion, thank you very much .

• Table 2 Attitude regarding VTE prophylaxis of medical staff in Guinea.

o These questions should NOT be combined to create a single overall affirmative response rate. Two of the questions are assessing the respondent’s assessment of their employers’ opinion of VTE prophylaxis; and two are multi part questions assessing concerns and difficulties where a negative response would be better assessed as a positive.

Response:

For the attitude regarding VTE prophylaxis, we provided the overall affirmative response rate, which could facilitate comparison with other similar studies. We also give the data for each question so that it is easy to analyze the knowledge blindness of the medical staff. This could be a reference for subsequent targeted training. Thank you for your suggestion.

• Table 3. Practice regarding VTE prophylaxis of medical staff in Guinea.

o Is it standard practice in Guinea for nurses to conduct VTE risk assessment? The discussion should reflect that. In the US, VTE risk assessment is usually conducted by clinicians.

o Question 4 is very vague and the much lower affirmative response rates may reflect that vagueness. Which VTE risk assessment scales are being referenced? In the US there are many different VTE risk assessment scales in practice, some are for different populations (cancer, pregnancy, pediatric) or different settings (medical, surgical, trauma) or developed in house versus externally validated. It is unreasonable to require providers in different specialties to have mastered ALL VTE risk assessment scales.

 Given the above issue, it is also recommended that an overall affirmative response rate not be reported.

Response: In Guinea, both nurses and doctors could perform the VTE risk assessment, according to your suggestion, we have described this in discussion part, thank you very much.

As you said, there do exist many VTE assessment scales, and our current survey did not restrict to a particular scale, which is one of our research shortcomings that we will describe within the research shortcomings. We provided the overall affirmative response rate, which could facilitate comparison with other similar studies. Thank you for your suggestion.

Discussion:

• Line 161: The authors discuss results that were not presented in the results section. But these are results I’m recommending they present, the breakout of topic areas for the knowledge questions.

Response: Thank you for your suggestions. We have described all these five parts of knowledge: the correct response rates for basic knowledge, risk assessment, basic prophylaxis, physical prophylaxis, and pharmacological prophylaxis were 70.814.4%, 65.016.5%, 72.322.3%, 44.719.8%, and 45.722.5%, respectively, with significant differences.

• Line 168: “found a positive attitude toward VTE prophylaxis among nurses” Should the word “more” been included? Also, must present the sub-analysis data in the results.

Response: Thank you for your suggestion, as shown in table 2, we have presented the sub-analysis data among nurses and doctors. Thank you very much.

• Line 171-174: Sentence needs adjustment to not make a causation statement.

Response: We have revised this sentence according to your suggestions, thank you.

• Line 174-176: Reference 12 is a review article and 13 occurs in Spain; neither address the statement as applied to “developing countries”.

Response: We have revised this sentence according to your suggestions, thank you.

• Line 178-181: Reference 14 presents the Padua Prediction Score and Reference 15 presents ASH’s guidelines for VTE among patients with cancer; at no time do either mention that conducting a risk assessment is mandatory for nurses but not clinicians. This is a disingenuous use of these references. If nurses conducting risk assessment is standard practice in Guinea then the authors should just state that fact.

Response: We have revised this sentence and the reference, thank you very much.

• Line 181-183: See previous comment regarding the vagueness of the question regarding VTE risk assessment “score”.

Response: We have described the different VTE risk assessment scores. Thank you very much.

Conclusion:

• Current conclusion is based on the current results. There may be a significant shift if authors undertake suggest revisions to methodology and analysis.

Response: We did not revise our result part, therefore, according to our present results we concluded that the medical staff’s knowledge level, attitude, and practice regarding VTE prophylaxis were generally poor. Thank you

Reviewer #2: Xinnong Liu et al report on "Knowledge, attitude, and practice regarding venous thromboembolism prophylaxis: A

multicenter cross-sectional study of medical staff in Guinea". The authors report on data from filling out self-designed questionnaire including four parts composed of demographic data, knowledge, attitude, and practice regarding VTE prophylaxis. The authors analysed 211 (86.1%).staff member responders' questionaires. "Correct" respnses were found in a range of 65% up to 84%, which was considered poor by the authors and the somewhat simple conclusion was to train better.

The topic is timely and of interest.

Emerging countries are cathching up and should be supported in their endeavour to train the staff in thrombosis prophylaxis.

Issues that should be considered:

1) 65%-84% positive answers should be wighed more differentiated. The (vast) majority seem to be compliant with the defined endpoints and answers correctly.

2) Given the situation in a emerging country, the verdict should be primarily positive.

3) The direct comparision of a partner hospital in China and /or of the western world would be desirable.

4) Data from the literature could alternatively flow in for more direct comparisions.

5) Consider also that ideal situations of procedures in emerging countries could differ greatly from other countries in the western world.

6) One example: "A multidisciplinary team must provide therapy to patients with VTE". This is how it works in other countries, but the structure of command may differ in Guinea.

7) "must" in VTE prophylaxis may induce counter actions and negative feelings towards outside authorities.

8) "What arrives" actually to the patients in need of VTE prophylaxis and to the ones who should not get it would be an important addition to the study, data may be available.

9) If not, recent data from other countries surveys on VTE prophylaxis could be directly compared.

10) More constructive advices on how to improve and perhaps to support the quality-measures would be very welcome, since this may apply to many other countries.

Response: Thank you for your suggestion. We all agreed with your suggestions. We have added comparison with other similar study in the discussion part. As you said, VTE prophylaxis is a multidisciplinary effort that also requires the involvement and coordination of hospital administrators. We have also described this in discussion part. Thank you very much.

6. PLOS authors have the option to publish the peer review history of their article (what does this mean?). If published, this will include your full peer review and any attached files.

Do you want your identity to be public for this peer review? For information about this choice, including consent withdrawal, please see our Privacy Policy.

Reviewer #1: No

Reviewer #2: No

---

## [Decision Letter · Decision Letter 1]

29 Dec 2024

PONE-D-24-29660R1Knowledge, attitude, and practice regarding venous thromboembolism prophylaxis: A multicenter cross-sectional study of medical staff in GuineaPLOS ONE Dear Dr. Liu,

Thank you for submitting your manuscript to PLOS ONE. After careful consideration, we feel that it has merit but does not fully meet PLOS ONE’s publication criteria as it currently stands. Therefore, we invite you to submit a revised version of the manuscript that addresses the points raised during the review process.

We look forward to receiving your revised manuscript.

Kind regards,

Mohammed Abutaleb, PhD

Academic Editor

PLOS ONE

Reviewers' comments:

Reviewer's Responses to Questions

**Comments to the Author**

1. If the authors have adequately addressed your comments raised in a previous round of review and you feel that this manuscript is now acceptable for publication, you may indicate that here to bypass the “Comments to the Author” section, enter your conflict of interest statement in the “Confidential to Editor” section, and submit your "Accept" recommendation.

Reviewer #3: All comments have been addressed

Reviewer #4: (No Response)

2. Is the manuscript technically sound, and do the data support the conclusions?

Reviewer #3: Yes

Reviewer #4: Partly

3. Has the statistical analysis been performed appropriately and rigorously? 

Reviewer #3: (No Response)

Reviewer #4: No

4. Have the authors made all data underlying the findings in their manuscript fully available?

Reviewer #3: Yes

Reviewer #4: Yes

5. Is the manuscript presented in an intelligible fashion and written in standard English?

Reviewer #3: Yes

Reviewer #4: Yes

6. Review Comments to the Author

Reviewer #3: thanks for the revision and address the all comments. Happy to be part to review the paper.

Reviewer #4: The study investigated the awareness of medical staff regarding venous thromboembolism (VTE) prophylaxis in Guinea. I recommend the following minor corrections:

1. Provide more details on how you determined the sample size of 245 cases. This could include the statistical methods or calculations used, any assumptions made, and the confidence level and margin of error considered.

2. Was the internal consistency of each part of the questionnaire assessed? Please report the Cronbach's Alpha value or any other relevant statistic that would help to understand the reliability of the questionnaire.

3. Clearly state the inclusion and exclusion criteria used in the study. This will help readers understand the characteristics of the study population and ensure the study's reproducibility and validity.

7. PLOS authors have the option to publish the peer review history of their article (what does this mean? ). If published, this will include your full peer review and any attached files.

**Do you want your identity to be public for this peer review?** For information about this choice, including consent withdrawal, please see our Privacy Policy .

Reviewer #3: **Yes: ** Mohammad Ashraful Amin

Reviewer #4: No

---

## [Author Response · Author response to Decision Letter 1]

6 Jan 2025

Response to reviewers

Reviewer #3: thanks for the revision and address the all comments. Happy to be part to review the paper.

Response: Thank you very much for your suggestions.

Reviewer #4: The study investigated the awareness of medical staff regarding venous thromboembolism (VTE) prophylaxis in Guinea. I recommend the following minor corrections:

1. Provide more details on how you determined the sample size of 245 cases. This could include the statistical methods or calculations used, any assumptions made, and the confidence level and margin of error considered.

Response: Thank you for your suggestion. The survey was conducted in all clinical departments of the three national hospitals, the names of the included departments have been described in the methodology section, and the 245 participants included all health care workers in the above departments. We surveyed all healthcare professionals in the above departments, totaling 245. It was not a sample survey. According to your suggestion, we have clearly described the inclusion and exclusion criteria of our study.

Thank you very much.

2. Was the internal consistency of each part of the questionnaire assessed? Please report the Cronbach's Alpha value or any other relevant statistic that would help to understand the reliability of the questionnaire.

Response: Thank you for your suggestions. According to your suggestion, we have analyzed the internal consistency of the questionnaire using the Cronbach's alpha analysis. The result showed that the Cronbach's alpha value of the questionnaire was 0.92. and the Cronbach's alpha analysis has been described in our manuscript.

Thank you very much.

3. Clearly state the inclusion and exclusion criteria used in the study. This will help readers understand the characteristics of the study population and ensure the study's reproducibility and validity.

Response: Thank you for your suggestion. According to your suggestion, we have described the inclusion and exclusion criteria of our study as following.

Inclusion criteria: The participants included all the clinicians, nurses, and pharmacists from all the clinical departments including internal medicine (Cardiology, Neurology, Acupuncture, Gastroenterology, Endocrinology, Respiratory medicine, and Nephrology), surgery (Neurosurgery, General surgery, Urology, Thoracic surgery, and Operating room), Intensive Care Unit ICU, and Emergencies.

Exclusion criteria: Administrators, Laboratory Physicians, Imaging Physicians, and Logistics Department staff.

Thank you very much.

---

## [Editor Report · Decision Letter 2]

11 Feb 2025

Knowledge, attitude, and practice regarding venous thromboembolism prophylaxis: A multicenter cross-sectional study of medical staff in Guinea

PONE-D-24-29660R2

Dear Dr. Liu,

We’re pleased to inform you that your manuscript has been judged scientifically suitable for publication and will be formally accepted for publication once it meets all outstanding technical requirements.

Kind regards,

Mohammed Abutaleb, PhD

Academic Editor

PLOS ONE
---

## [Editor Report · Acceptance letter]

PONE-D-24-29660R2

PLOS ONE

Dear Dr. Liu,

I'm pleased to inform you that your manuscript has been deemed suitable for publication in PLOS ONE. Congratulations! Your manuscript is now being handed over to our production team.

Kind regards,

on behalf of

Dr. Mohammed Abutaleb

Academic Editor

PLOS ONE